# The Non-Nutritional Factor Types, Mechanisms of Action and Passivation Methods in Food Processing of Kidney Bean (*Phaseolus vulgaris* L.): A Systematic Review

**DOI:** 10.3390/foods12193697

**Published:** 2023-10-09

**Authors:** Zifan Zhang, Chunxiu Liu, Sisi Wu, Tiezheng Ma

**Affiliations:** 1School of Food and Health, Beijing Technology and Business University, Beijing 100048, China; 2Beijing Higher Institution Engineering Research Center of Food Additives and Ingredients, Beijing Technology and Business University, Beijing 100048, China

**Keywords:** kidney bean, non-nutritional factors, action mechanism, passivation method

## Abstract

Kidney beans (KBs), as a traditional edible legume, are an important food crop of high nutritional and economic value worldwide. KBs contain a full range of amino acids and a high proportion of essential amino acids, and are rich in carbohydrates as well as vitamins and minerals. However, KBs contain a variety of non-nutritional factors that impede the digestion and absorption of nutrients, disrupt normal metabolism and produce allergic reactions, which severely limit the exploitation of KBs and related products. Suppressing or removing the activity of non-nutritional factors through different processing methods can effectively improve the application value of KBs and expand the market prospect of their products. The aim of this review was to systematically summarize the main types of non-nutritional factors in KBs and their mechanisms of action, and to elucidate the effects of different food processing techniques on non-nutritional factors. The databases utilized for the research included Web of Science, PubMed, ScienceDirect and Scopus. We considered all original indexed studies written in English and published between 2012 and 2023. We also look forward to the future research direction of producing KB products with low non-nutritional factors, which will provide theoretical basis and foundation for the development of safer and healthier KB products.

## 1. Introduction

Edible legumes such as soybeans, peanuts, peas, broad beans and kidney beans (KBs) are an important part of the human diet because they are a good source of protein, minerals, vitamins and bioactive compounds [1]. KBs (*Phaseolus vulgaris* L.), an essential legume in human nutrition due to their high nutritional value, are among the most widely cultivated and consumed legumes in the world. They have been reported to enhance metabolism, alleviate chronic diseases, and boost immunity. They are native to the Americas and widely cultivated in tropical and subtropical countries in Europe, Africa and Asia, and have become one of the main sources of plant protein and dietary starch in the daily diet of the residents [2,3,4]. KBs are rich in carbohydrates, protein, dietary fiber, minerals and vitamins, are gluten-free and lower in fat than other common edible beans and contain highly active glycosidase hydrolase, which can prevent starch hydrolysis, reduce glucose absorption and reduce fat production [5,6,7]. The nutritional value of KBs is attributed to their high amino acid content, which aligns with the reference protein model proposed by FAO/WHO [8,9]. KBs also have excellent nutritional value, making them an effective option for improving nutritional intake and reducing nutritional deficiencies in economically restricted areas, or as a high-quality protein source for vegetarians.

However, legumes, including KBs, are underutilized, in part because their consumption can lead to flatulence and lower protein digestibility. The reduced protein digestibility is attributed to the presence of natural barriers and non-nutritional components within the plant cells [10]. Numerous secondary metabolites naturally synthesized in high concentrations during the growth of KBs can produce adverse reactions upon consumption and reduce the nutrient utilization of human food and/or food intake, collectively referred to as non-nutritive compounds [11]. In daily life, the consumption of unprocessed and improperly processed KB products is very likely to cause food poisoning. As research on non-nutritional factors has advanced, studies have reported that many compounds found in lower concentrations, often referred to as plant bioactive components or anti-nutritional factors, can have beneficial effects in preventing coronary atherosclerotic disease and various cancers [12,13,14]. The existence of non-nutritional factors such as toxic proteins and toxic alkaloids reduces the nutrient absorption rate of KBs and KB products, affecting the health of humans and animals [15]. The non-nutritional factors in KBs mainly include lectin, trypsin inhibitor, phytic acid and saponin, etc., which directly inhibit protease activity, chelate ionic cofactors or form irreversible complex to block protein digestion [16,17]. These non-nutritional factors affect the absorption of nutrients by humans and animals, interfere with their normal metabolism, cause allergic reactions in human body and cause adverse reactions including skin, gastrointestinal and respiratory, among others, and cause food poisoning in serious cases. Therefore, non-nutritional factors have caused the decline of the quality and nutritional value of KB products, which has restricted the further development and utilization of KBs and their products by humans [18,19]. Earlier research on non-nutritional factors focused on soybeans, broad beans and other legume feedstuffs because they can affect nutrient absorption in animal feed. KBs, as one of the main sources of plant protein, are increasingly valued by consumers, and are increasingly utilized in modern diets and food processing, resulting in increasingly frequent food safety incidents, and it is urgent to pay attention to their dietary safety. Hence, it is important to eliminate or weaken the non-nutritional factors in KBs in food production and processing. At present, the research on the non-nutritional factors in KBs mainly focuses on the allergenicity and elimination of KB lectins, but there are few reports on the action mechanism of KB non-nutritional factors, their inactivation and related analysis in food processing.

Therefore, this paper reviews the types, characteristics and action mechanism of non-nutritional factors in KBs, elucidates the effects of intake of non-nutritional factors on human health and focuses on the specific inactivation methods of various types of non-nutritional factors in food processing. Selecting appropriate passivation methods based on the characteristics of various non-nutritional factors, and passivating or eliminating them while ensuring that other nutrients are not destroyed, could provide a certain theoretical basis for promoting the consumption of KBs and the development and utilization of KB-based foods.

## 2. Materials and Methods

### 2.1. Data Search and Study Selection

Articles for full-text review were selected by screening the titles and abstracts of all publications yielded by the systematic search of Web of Science, PubMed, ScienceDirect and Scopus. Three combinations of the following research terms were used in the search engines: (i) (legume) and (antinutritional factors OR non-nutritional factors); (ii) (kidney bean OR kidney beans) and (antinutritional factors OR non-nutritional factors) and (passivation method OR removal method); (iii) (kidney bean protein) and (antinutritional factors OR non-nutritional factors) and (mechanism of action) and (passivation OR removal method in food processing). All original indexed studies published in English between 2012 and 2023 were included in this review. Abstracts, scientific opinions and papers without full text were excluded. The main objective of the selected articles was to investigate non-nutritional factors in KBs. In addition, it is important to emphasize that clarifying the types of non-nutritional factors in KBs and exploring their action mechanisms, as well as specific methods of passivation in food processing, are the main issues that need to be addressed in a systematic review.

### 2.2. Data Extraction

The methodology used in this review is based on the Preferred Reporting Items for Systematic Reviews and Meta-Analyses Guidelines (PRISMA). This research tool consists of a checklist and a four-stage flowchart to assess randomized studies but can also be used as a basis for reporting systematic reviews of other types of studies [20]. Articles were first selected based on title and abstract information, and then chosen after reading the complete available documentation without meta-analysis.

## 3. Results

### 3.1. Eligibility of Studies

The flowchart of the systematic review based on the PRISMA guidelines is presented in Figure 1. From a total of 4989 titles and abstracts, 283 articles were selected for full-text review. The high number of excluded articles can be attributed to several factors: repetition (n = 485), the absence of the main inclusion criterion, which is ‘non-nutritional factors in kidney bean’ or ‘antinutritional factors in kidney bean’ in the titles or abstracts (n = 3123), articles that lacked full text (n = 689) and publication languages other than English (n = 409). Among the 283 articles, 55 were excluded due to insufficient detailing of the specific mechanisms of action of non-nutritional factors; 30 were excluded due to the lack of methods for blunting or eliminating specific types of non-nutritional or antinutritional factors and an additional 75 were excluded because they were published before 2012. For this review, 123 articles were selected: 17 of them identified the types of non-nutritional factors in KBs and summarized their properties, 41 identified the action mechanisms and health effects of each type of non-nutritional factor, and 65 explored methods for the passivation or elimination of non-nutritional factors in food processing. From this selection process, three main points were identified: the types and properties of non-nutritional factors in KB; the specific action mechanisms of each type of non-nutritional factor and the specific passivation methods for non-nutritional factors of different characteristics.

### 3.2. Types and Action Mechanism of Non-Nutritional Factors in KBs

The presence of non-nutritional factors in KBs varies in form and properties, and has different effects on the body. On the one hand, these non-nutritional factors include lectins, which bind carbohydrate residues and cause allergic reactions, trypsin inhibitors, which bind proteases and inhibit the activity of related proteases, thereby affecting the digestion of starch and protein, and phytates and tannins, which bind minerals and proteins to affect their bioavailability and digestibility, respectively [21]. The consumption of raw or improperly processed KBs can affect the absorption and utilization of other nutrients, accompanied by decreased protein digestibility, and even symptoms such as loss of appetite, weight loss, dyspnea and even poisoning [22]. On the other hand, these non-nutritional factors, such as lectins, trypsin inhibitors, polyphenolics and saponins, also serve as defense mechanisms to protect the plant from herbivores and pathogens. Additionally, they act as signaling molecules in the interaction between the plant and its biological environment [23]. The existing forms and properties of non-nutritional factors in KBs are shown in Table 1.

#### 3.2.1. Lectin

KB lectin is an oligomeric globulin consisting of three polypeptide subunits (*α*-, *β*- and *γ*-subunits) with a molecular weight range of 43 kDa to 53 kDa [29]. Lectin is the main non-nutritional factor in KBs, and it has very high hemagglutination activity and can form a three-dimensional structure when it reacts with glycoproteins and carbohydrates [30,31]. The consumption of raw or improperly heated KBs can easily cause food poisoning, while the thorough cooking of soaked KBs (95 °C, 1.0 h) can lead to complete inactivation of or reduction in lectins below the limit of detection. Soaking can significantly reduce the level of lectins, making the cooking process more effective at inactivating lectins by up to 99% [32,33]. The distribution of lectins in different Brassica plants is very different, and raw KB seeds are rich in lectins, so the lectins extracted from KB seeds can be used commercially [34]. In addition, lectin concentrations decrease as fresh KBs mature. Compared to soybeans (*Glycine max* (L.) Merr.) and peas (*Pisum sativum* L.), red KBs have the highest concentration of lectins, with an average of 20,000–70,000 hemagglutination units (hau/g) in raw beans [35]. Lectins are proteins that act as antigens to induce the production of specific antibodies in vivo; therefore, based on the high specificity of molecular recognition between antibodies and antigens, the allergenicity of lectins in natural and processed KBs can be analyzed via quantitative enzyme-linked immunosorbent assay (ELISA) [36]. Based on the Clostridium perfringens phytohemagglutinin structural template, the three-dimensional structure of black KB lectin was modeled, and lymphocyte epitopes were screened and identified with immunoinformatic, followed by ELISA, lymphocyte proliferation and cytokine profiling analysis were validated and epitopes in the allergenic protein were identified, leading to the diagnostic and therapeutic approaches for lectin allergy [37].

Lectins are glycoproteins that cannot be hydrolyzed by proteolytic enzymes in the digestive tract. Due to the relative resistance of allergen proteins to proteolytic enzymes, the epitope region of undigested lectin can bind to intestinal epithelial cells and cause border damage to the fine intestinal epithelium [38]. Lectins stimulate antibody production and induce T-cell immunity across the epithelial barrier while being able to interact with digestive enzymes leading to nutrient deprivation, IgE-mediated immediate hypersensitivity and non-IgE-mediated reactions (cell-mediated immunity), eliciting clinical symptoms of nausea, vomiting, diarrhea and other digestive disorders [39,40,41]. The negative effects of KB lectin have been confirmed in mammals using rats as a model. After feeding rats with KB lectin, the binding of the lectin to specific carbohydrate structures in the brush border of the neonatal rat intestinal epithelium significantly blocks the ability of the rat intestinal epithelium to absorb endocytosis, including receptor-mediated bovine IgG and more nonspecific human serum albumin [42]. Similarly, Nciri et al. [43] found that the lectin content in the serum of rats fed with white KB lectin rose to 3% after 10 h, indicating that white KB lectin can easily cross the intestinal barrier into the blood, disrupt normal cell function and cause immune disorders. The sensitization mechanism of KB lectin is depicted in Figure 2.

There has also been extensive recent research on the potential health benefits of lectins. KB lectins may have the potential to prevent tumor formation by inducing tumor cell death through enhanced apoptosis [44]. Additionally, KB lectins can recognize microorganisms, detect changes in carbohydrates on the surface of immune cells and subsequently interact to trigger signal transduction, leading to the production of cytokines. This, in turn, exerts immunomodulatory activity and induces effective immune responses against microbial infections or tumors [45]. Therefore, lectins with immunomodulatory activity have a wide range of potential medicinal uses and may offer new strategies for developing drugs with both immunomodulatory and direct antitumor effects.

#### 3.2.2. Trypsin Inhibitor

Trypsin inhibitors are antimetabolic proteins associated with the inhibition of digestible enzymes that bind to trypsin and chymotrypsin, inhibiting protein digestion and impeding amino acid absorption [25]. These inhibitors are usually thermally unstable, so their activity depends on the temperature and time of cooking. Using a multi-step strategy, trypsin inhibitors can be isolated and purified from KBs with a recovery rate of 15% and a purification factor of 39.4. The obtained product can maintain inhibitory activity in the pH range of 3–11 and temperature range of 40–60 °C [46]. Similarly, trypsin inhibitors in white and red KBs were isolated through heating precipitation, and the recoveries of trypsin inhibitors obtained via incubation at 70 °C for 10 min reached 156.8% and 252.8%, respectively. When supplemented with ammonium sulfate at a concentration of 60% to 80%, the purity of the product obtained from white and red KBs increased by 41-fold and 88-fold, respectively, and the percentage of activity increased by 26% and 126%, respectively [47]. Different genotypes and cultivation conditions greatly affect the level of non-nutritional factors in KB seeds. By comparing the changes in non-nutritional factors in KBs with different genotypes and cultivation conditions, it was found that trypsin inhibitors were most affected by cultivation conditions, and stress conditions significantly increased the expression of trypsin inhibitors in the vegetative and reproductive stages of KBs by 46.2% [48].

The antinutritional effect of trypsin inhibitory factor is mainly manifested in the following two aspects: first, it combines with trypsin in the small intestinal fluid to generate inactive complexes, which reduces the activity of trypsin, resulting in a decrease in protein digestibility and utilization; second, it leads to the depletion of the animal’s endogenous protein [49]. The decrease in trypsin content in the small intestine due to the combination of trypsin with trypsin inhibitor and excretion through the feces stimulates the increased secretion of cholecystokinin, which leads to the increased secretion of enteroglucagogue peptide and feedback causes pancreatic hyperfunction, prompting the pancreas to secrete more trypsinogen into the intestine [50]. The excessive secretion of trypsin leads to pancreatic hyperplasia and hypertrophy, causing the dysfunction of digestion and absorption, and severe diarrhea. Since trypsin is particularly rich in sulfur-containing amino acids, a large amount of compensatory secretion of trypsin leads to the endogenous loss of sulfur-containing amino acids in the body, thereby exacerbating the imbalance of amino acid metabolism and even leading to the occurrence of diseases [51].

On the other hand, there were also studies showing that trypsin inhibitors were effective in preventing pancreatic injury caused by the premature activation of trypsin, suggesting a potential role in pancreatitis treatment [52]. Sibian and Riar [53] found that trypsin inhibitors, when consumed in appropriate amounts, had significant effects on certain types of cancer, Alzheimer’s disease, autoimmune diseases, fungal infections and inflammatory diseases.

#### 3.2.3. Phytic Acid

Phytic acid or phytate is the major storage form of naturally occurring phosphate in KBs, a source of mineral nutrients and inositol used during germination, and is capable of chelating mineral cations and proteins, forming insoluble precipitation complexes, reducing the bioavailability of minerals and protein digestibility in the gastrointestinal tract [54,55]. KB seeds are rich in magnesium, calcium, iron and zinc, but also contain high levels of phytic acid, which limits the absorption of minerals during digestion [56]. The phytic acid content of different bean species varied greatly, and the content of phytic acid extracted from beans ranged from 119.82 to 149.30 mg/100 g. The phytic acid content of red KBs extracted using high-performance liquid chromatography and ion exchange chromatography reached 149.30 mg/100 g [57]. This is the reason for the low bioavailability of Fe, Zn and Ca in the body after the consumption of KBs due to phytic acid binding to minerals.

Phytates can interfere with mineral levels in organisms and can also negatively affect the bioavailability of other nutrients, including proteins. In addition, phytates can negatively affect the activity of digestive enzymes such as carboxypeptidases and aminopeptidases through the chelation of mineral cofactors or interactions with proteins as enzymes or substrates [58,59]. The phytic acid content of KBs ranges from 0.2 to 2.9%, which can absorb more than 50% of the minerals in food [60]. The structure and action mechanism of phytic acid in KBs is shown in Figure 3.

However, the chelating properties of phytic acid, which binds to minerals and proteins, have the potential to offer various health benefits, including a reduced risk of diabetes and certain malignancies, improved heart health and the prevention of kidney stone formation [61]. Phytates indeed serve as antioxidants and can regulate excess heme iron while reducing advanced glycosylation end products in patients with type II diabetes [62]. Standardized phytate dosages, determined through in vitro and in vivo studies, can be instrumental in the development of novel drugs.

#### 3.2.4. Tannin

Tannins are water-soluble phenolic metabolites that can be classified into three categories: hydrolysable tannins (e.g., galliotannins and ellagitannins), condensed tannins (e.g., proanthocyanidins) and complex tannins. Condensed tannins can bind and precipitate protein fractions, making them difficult to digest and absorb and thus limiting their availability [27,63]. The tannins that exist in large quantities in KBs belong to condensed tannins, and are mainly located in the seed coat of the beans [64]. Different KB varieties have different tannin levels. A comparative study of tannin content in black KBs, light-spotted KBs, crimson KBs, large white KBs, red-spotted KBs, small red KBs and small white KBs showed that the tannin content of colored beans (black, light-spotted, crimson and red-spotted) was 13.0 to 19.9 g/kg, while the tannin content in white KBs was about 0.03 g/kg [65]. Similarly, a comparative study of the tannin content of red KBs, yellow KBs and black and white spotted beans showed that red KBs had the highest tannin content of 288.6 mg/100 g [66]. The easy-to-cook (ETC) and hard-to-cook (HTC) properties of KBs have also been reported to affect the tannin content, ranging from 0.03 to 1.26 mg/g for ETC seeds and 0.03 to 1.18 mg/g for HTC seeds [67].

Tannins inhibit hydrolytic enzymes such as trypsin, alpha-amylase, glucoamylase and lipase, and bind not only to proteins but also to minerals and vitamins, making them unusable [68]. Tannins form complexes with proteins through hydrogen bonds through hydroxyl and carbonyl groups, which affect the digestibility of proteins and lead to the inhibition of the utilization of essential amino acids and minerals [69]. Ravoninjatovo et al. [70] showed that soaking treatment accelerated the leaching of tannins in KBs, and about 70% of the tannins were lost after soaking for 10 h. The specific action mechanism of tannins in KBs is shown in Figure 4.

Like most phenolic compounds, tannins exhibit a wide range of biological activities, including cardioprotective, anti-inflammatory, anticancer, antiviral and antimicrobial properties, primarily due to their antioxidant and anti-free-radical activities [71]. In moderate amounts, tannins have the potential to serve as therapeutic agents against a wide range of diseases. Furthermore, as polyphenols, tannins play a crucial role in the antioxidant activity of plants, serving to protect crops from pests during their growth [72].

#### 3.2.5. Saponin

Saponins are triterpenoids or steroidal glycosides found in KBs that affect nutrient absorption by inhibiting some metabolic or digestive enzymes and binding to essential nutrients such as iron, zinc and vitamin E [28]. Saponins are mainly found in KB hulls, and Nguyen et al. [73] found that the content of saponin in the hulls of large white KBs was four times higher than that of peeled large white KB seeds. Saponins are thermally unstable, and the degradation of saponins αg and βg during heat treatment increased the concentration of saponins Ba and I, and these decreases may be related to the production of maltol through saponin degradation [74].

The consumption of saponins often results in the breakdown of red blood cells, limiting protein digestion and vitamin and mineral absorption, leading to hypoglycemia [75]. Saponins can also attach to intestinal cells and interfere with the nutrient absorption of the intestinal membrane, causing diarrhea and vomiting, and in severe cases, leaky gut [76]. In addition, some saponins can form insoluble complexes with cholesterol, preventing its absorption from the small intestine [77]. Similar to tannin, soaking treatment can also significantly reduce the content of saponin in KB seeds, and the loss of 68.8% after soaking [78].

Nevertheless, saponins offer significant health benefits due to their triterpenoid glycosidic elements, which are linked to one or more oligosaccharide groups, enabling them to absorb free radicals and activate antioxidant enzymes [12]. Terpenoids, a sub-group of triterpenoids, have also been associated with various health benefits, including the reduction in harmful plasma cholesterol levels and the demonstration of antioxidant, anticancer, and antimicrobial properties [79]. Currently, there is a relatively limited body of research on KB saponins, making them a prominent topic for future studies on non-nutritional factors found in edible legumes, including common beans.

### 3.3. Control Measures of Non-Nutritional Factors in Food Processing

Various processing techniques, such as soaking, autoclaving, dehulling, boiling, sprouting, fermentation and ultrasonication, all attempt to enhance the potency of beans and increase their nutritional value and digestibility [80]. KB non-nutritional factors can be divided into two categories according to their heat sensitivity: heat-sensitive non-nutritional factors and heat-stable non-nutritional factors [81]. Among them, trypsin inhibitor and lectin are heat-sensitive, while saponins, tannins and phytic acid are heat-stable [82]. For heat-sensitive non-nutritional factors, heat treatment can significantly reduce the activity of non-nutritional factors in KBs, while heat-stable non-nutritional factors need to be inactivated through non-heat treatment.

According to the properties of KB non-nutritional factors, heat treatment or non-heat treatment can be selected as the passivation method [83]. The process for passivating non-nutritional factors in KBs and the development of KB products with low non-nutritional factors is shown in Figure 5. The heat treatment method is simple and easy to implement, and has been widely used in actual production and processing, but its energy consumption is relatively high, and other co-existing bioactive compounds are also easily inactivated during the process. Non-heat treatment includes chemical treatment, enzyme treatment, fermentation treatment and other methods [84]. Among them, chemical treatments are mostly used in the feed industry rather than the food industry. This is because some chemical substances introduced by the treatment will seriously affect the taste and quality of the product and reduce the palatability, and the waste liquid produced will cause a large amount of pollution to the environment [85,86].

#### 3.3.1. Elimination of Heat-Sensitive Non-Nutritional Factors

Moderate heat treatment will weaken the non-nutritional factors contained in the KBs, gelatinize the carbohydrate components to facilitate digestion and improve the utilization of essential amino acids [32]. The heating method is efficient, simple and low-cost, and has no residue problems, but it is only suitable for heat-unstable non-nutritional factors such as lectins and trypsin inhibitors, and is invalid for heat-stable non-nutritional factors such as phytic acid and saponins [69]. Insufficient heating during the heating process will not eliminate heat-sensitive non-nutritional factors, while excessive heating will destroy arginine, lysine and sulfur-containing amino acids [87].

(1) Boiling treatment

Boiling treatment is one of the more commonly used methods to eliminate non-nutritional factors in KBs. Khrisanapant et al. [88] studied the boiling treatment of cowpea, chickpea and KB, and achieved the inactivation of trypsin inhibitor and lectin in KB through heat treatment, improving the rate and degree of starch digestion of the three legumes. The effect of heat treatment inactivating non-nutritional factors was closely related to the type of KB and the rate of boiling. Wiesinger et al. [89] found that after boiling, the phytate concentration in white-KB-based spaghetti was reduced from 12.90 mg/g to 9.25 mg/g compared to the whole bean, which means that the milling can assist boiling to promote the decomposition of phytic acid, thereby promoting the absorption of dietary iron by humans. Similarly, the combination of hull processing and soaking can improve the relative water absorption of KB seeds, resulting in a significant reduction in polyphenol and tannin content and trypsin inhibitor activity in the boiled product [90]. Wiesinger et al. [91] also found that boiling speed affects the amount of phytic acid retained in KBs, since fast-cooking products in the yellow, cranberry and red KBs had higher phytic acid density and retained values compared to medium- and slow-boiling products.

(2) Roasting treatment

The high temperature generated during roasting can inactivate non-nutritional factors, and compared with the boiling treatment, the roasting treatment takes less time to inactivate due to the higher temperature. Godrich et al. [26] found that the roasting treatment reduced the phytic acid content of red KBs by 16.8%, while this content did not change significantly after soaking and boiling treatments. Khattab and Arntfiel [92] also reported that the roasting treatment reduced the phytic acid content of Canadian and Egyptian KBs by 40.2% and 36.0%, respectively, which was comparable to micronization. The decrease in KB-derived phytic acid content during the roasting treatment may be due to the formation of insoluble complexes with certain minerals [93,94].

(3) Microwave treatment

Microwave treatment, like baking and boiling, can remove non-nutritional factors in KBs, and the combined use of different heat treatment methods can have superimposed effects. Khattab and Arntfiel [92] found that the microwave treatment had a higher effect on the degradation of tannin and phytic acid in Canadian and Egyptian KBs than the roasting or boiling treatments. Li et al. [18] combined microwave and hot air drying to pre-paste KBs, removing free water from the KBs and gelatinizing the starch on their surface, thereby destroying the dense structure of KBs and degrading the lectin in them.

#### 3.3.2. Elimination of Heat-Stable Non-Nutritional Factors

Legumes usually need to be properly processed before consumption to optimize the edible quality and reduce or eliminate metabolic disorders caused by non-nutritional factors. In general, cooking and heat treatment often lead to a decrease in the nutritional quality and content of bioactive compounds in foods, but also increase palatability [95]. However, for heat-stable non-nutritional components such as phytic acid, tannins and saponins, non-thermal food processing methods need to be selected based on their chemical structure, distribution in seeds, biological effects and their solubility. Processing techniques such as dehulling, soaking, sprouting, chemical treatment and fermentation are effective methods to reduce or remove these non-nutritional factors [96].

(1) Dehulling treatment

Removing the seed coat of legumes is a common method to enhance palatability and remove non-nutritional factors, although it results in a partial loss of minerals and dietary fiber [96]. Nakitto et al. [97] found that the use of a combination of dehulling and roasting or boiling treatments was effective in reducing tannins and phytic acid below the limits of detection, whereas 95.7% of tannins and 81.7% of phytic acid residues remained in unhulled KBs after roasting, and 68.9% of tannins and 53.9% of phytic acid residues remained after boiling. Anino et al. [66] prepared a KB-based beverage by soaking, dehulling and heat treating three types of KBs and found that the phytic acid content of the products based on red, yellow and purple KBs was reduced by 77.6%, 70.5% and 80.8%, respectively, and the authors concluded that the rate of phytic acid removal showed a positive correlation with the phosphorus content of the beans.

(2) Soaking treatment

Soaking is a common process for processing edible beans, usually at room temperature, and the choice of solvent and the duration of soaking affect the elimination effect of non-nutritional factors. Yasmin et al. [98] soaked red KBs in water, citric acid and sodium bicarbonate solutions, respectively, and found that the content of tannin remained unchanged after soaking in aqueous solution, while soaking in citric acid and sodium bicarbonate solutions could effectively reduce the content of tannin, and phytic acid remained unchanged after different soaking treatments. Zhu et al. [99] soaked red KBs in CaCl_2_ solution and found that the migration of exogenous calcium and its enrichment in cotyledons contribute to the activation of phytase, thereby greatly accelerating the hydrolysis of phytic acid in KBs. Haileslassie et al. [100] treated red KBs with either soaking or cooking after germination, and observed that red KBs subjected to 72 h of germination followed by steaming showed a significant reduction in both tannin and phytate content, with reductions of 22.5% and 39.0%, respectively.

(3) Germination treatment

The germination of legumes is one of the outstanding green food development techniques due to the activation of endogenous enzymes, such as protease and amylase, during the germination process to promote the degradation of proteins and carbohydrates, which can reduce the content of non-nutritional factors and digestive inhibitors, increase the utilization rate of proteins and minerals and enhance nutrient accessibility [101,102]. Nciri et al. [103] found that the content of lectins increased slightly during the first 4 days of the germination of KBs, while it decreased to less than one-tenth of the initial content after 9 days. Different varieties of KBs were treated with germination and its antinutrient content varied significantly. Owuamanam et al. [104] germinated red KBs and found that the content of lectins, trypsin inhibitors and phytic acid in KBs decreased by 84.2%, 72.1% and 69.3%, respectively, while the mineral content increased by 16.7%.

Similar experimental results were obtained by Sibian and Riar [53], who found that trypsin inhibitor, phytic acid and tannin content were reduced by 45.24%, 36.23% and 33.63%, respectively, after the germination treatment of KBs, which was attributed to the ability of the germination process to activate enzyme systems such as phytase. The use of different elicitors during the germination treatment can affect the rate of degradation of non-nutritional factors in KBs. Limón et al. [105] used different elicitors such as ascorbic acid, folic acid or glutamic acid to treat KBs, followed by germination, and found that the folic acid-treated samples had the relatively fastest rate of degradation of lectins, and lectins in the sample were almost completely degraded after 8 days of germination, irrespective of the elicitors used.

Soaking KBs prior to germination can significantly enhance the passivation effect of non-nutritional factors. Mugabo et al. [106] found that pre-soaking KBs in distilled water for 24 h prior to germination reduced the content of tannins, phytic acid, lectins and trypsin inhibitors by 94.5%, 79.3%, 80.3% and 77.7%, respectively.

(4) Chemical treatment

Low-pH-induced changes can cause the blunting of non-nutritional factors and the elimination of allergenic components in KBs. Zhao et al. [107] found that the gradual unfolding and dissociation of lectin protein in black KBs under low acidic conditions could terminate the surface exposure of epitopes and lead to a loss of surface accessibility and flexibility for IgE recognition, with the relative IgE binding capacity decreasing by 59.07% at pH 1.0. Sun et al. [108] used a combination of low-pH-induced changes and heat treatment to reduce the lectin content in black KBs, which reduced the allergenicity of black KBs by altering the secondary and tertiary structure of the proteins, resulting in a loss of availability in lectin conformational or linear epitope, and reduced the IgE binding capacity by 63.28%.

The addition of some chemicals can also modify the protein fractions in KBs to reduce their non-nutritional factor content. He et al. [109] chemically treated black KBs with polyethylene glycol and found that polyethylene glycol derivatives could be linked to target proteins through covalent linkage at the terminal N of lysine residues, and the resulting shielding effect of the protein side chains could block the biologically active structural domains of the target proteins and reduce protease cleavage or glycoprotein recognition, which would decrease the hemagglutination activity by at least 75%. Yang et al. [110] also chemically treated black KBs with methoxy polyethylene glycol derivatives, which shielded a large area around the protein molecule, thereby obstructing the access of antibodies or sensitized lymphocytes to the protein epitopes, and reduced allergic reactions induced by lectins by 55.4%.

(5) Enzymatic treatment

Similar to chemical treatments, enzyme treatments can also act specifically on non-nutritional factors and change their molecular structure, thus effectively reducing the allergenicity of KBs and preparing hypoallergenic KB products. Israr et al. [111] added exogenous phytase to red KBs to reduce the concentration of phytate and soluble oxalate, and the concentration of phytate decreased by 57% after 30 min of enzymatic action. Saad et al. [112] hydrolyzed black, white and red KBs using Alcalase^®^, which can hydrolyze several proteins to small peptides, and found that the lectin content in all three types of beans decreased significantly with increasing hydrolysis time. Kasera et al. [113] compared the effect of Alcalase^®^ and Flavourzyme^®^ on the degradation of non-nutritional factors in KBs and found that hydrolysis using Flavourzyme^®^ reduced IgE binding in KBs by 62%, which was superior to that of Alcalase^®^ (43%) because different hydrolysates may block different antigen binding sites on the IgE, thereby acting as the natural inhibitor for allergic reactions. Enzyme treatment maximizes the retention of beneficial nutrient activity while reducing non-nutritive factors. AL-Ansi et al. [114] successfully reduced lectin activity to 6.97% while maintaining 85.89% of α-amylase inhibitor activity through the acid protease treatment of KB extract. This KB extract reduces starch hydrolysis and effectively lowers the glycemic index of common staple foods such as bread, rice, steamed bread and cornbread.

#### 3.3.3. Fermentation Treatment

Fermentation, similar to enzymatic digestion, is a method to specifically degrade non-nutritional factors in legumes, and it has been shown to increase the activity of phytase, degrading and inactivating mineral-complexed phytic acid [115]. The process of fermentation improves the digestibility of starches and proteins as well as the bioavailability of minerals in legumes [116]. Fermentation can be divided into solid-state fermentation and liquid-state fermentation (or submerged fermentation). In solid-state fermentation, microorganisms are grown on a solid substrate surrounded by a continuous gas phase, the space is filled with gas, the moisture content during fermentation is in the range of 12–70%, and the strains used are usually adapted to a natural growing environment, where the mycelium can grow in the interstices of the solid particle substrate [117,118]. In submerged fermentation, microorganisms are able to grow in an environment with high levels of free water containing nutrients and rapidly consume substrates [110]. Therefore, legume fermentation is more often carried out under solid-state conditions using fungi, whereas under submerged fermentation it is carried out using bacteria. The specific forms of action in submerged and solid-state fermentation processes are shown in Figure 6.

(1) Submerged fermentation

Submerged fermentation treatment can particle-blunt the non-nutritional factors and thus enhance the absorption of nutrients from KBs. Chaturvedi and Chakraborty [119] used probiotic *Lactobacillus casei* for the fermentation treatment of red KBs to produce a beverage product; the non-nutritional components of phytic acid, tannins and saponins in KBs were reduced by 29%, 58% and 28%, respectively. Subsequently, Chaturvedi and Chakraborty [120] switched to probiotic *Lacticaseibacillus casei* ATCC 335 and fermented red KBs for the production of beverage products and the fermentation reduced phytic acid, tannins and saponins in the KBs by 53%, 46% and 44%, respectively. The reduction in the content of non-nutritional components after the fermentation treatment can effectively reduce the occurrence of digestive disorders such as bloating and flatulence.

(2) Solid-state fermentation

Solid-state fermentation is more commonly used in food legume processing than submerged fermentation, since fungi are used more in legume fermentation than bacteria, and fungal fermentation can enhance the functionality and nutritional properties of KBs as food products. The fermentation of KBs using *Pleurotus ostreatus* by Espinosa-Páez et al. [121] resulted in a 39% reduction in tannin content, which was mainly through the action of the tannase enzyme during the fermentation process, and the digestibility of the fermented proteins increased from 44.06% to 69.01% due to the reduction in non-nutritional factors. Subsequent studies by Espinosa-Páez et al. [122] have further demonstrated that non-nutritional factors such as phytic acid and tannins reduce protein bioavailability, whereas fermentation treatments using *Pleurotus ostreatus* are able to increase the content of essential amino acids in KBs while degrading non-nutritional factors. Lactic acid bacterial strains favored the removal of trypsin inhibitors and tannins from gluten-free KB dough through fermentation treatment and increased the amino acid and phenolic content as well as the antioxidant activity of the KB doughs, as found by Sáez et al. [123], who used *Enterococcus durans* CRL 2178 and *Weissella paramesenteroides* CRL 2182 to reduce the trypsin inhibitor and tannin content of fermented KB doughs by 85.1% and 95.6%, respectively.

## 4. Conclusions and Prospective Research

KBs as a food ingredient are currently used mainly in meat products such as sausages, but are rarely used in other foods. This is mainly due to the presence of allergenic components such as lectins and anti-digestive components such as trypsin inhibitors, tannins, saponins and phytic acid in KBs, which need to be processed using physical, chemical and biological treatments to ultimately quench these non-nutritional components. Extensive research has been conducted on the nutritional, functional and protein modification and processing applications of KB proteins, but relatively limited research has been conducted on the types, action mechanisms and passivation methods of its non-nutritional factors. Currently, the main method for eliminating non-nutritional factors from KBs is heat treatment; the high temperatures generated during heat treatment can inactivate the non-nutritional factors before they become harmful to humans; however, the denaturation of KB protein during heat treatment, as well as the inactivation of some thermally unstable nutrients, will lead to a decrease in the nutritional value of KBs. Therefore, there is a need to reduce the allergenicity and quench the anti-digestive components of KBs through non-heat treatments such as glycosylation, fermentation or enzymatic treatments so as to ensure the nutritional value of KBs and further promote their promising applications in the food industry.

Although a number of generally applicable passivation methods for the non-nutritional factors of edible legumes have been developed, the production of high-value-added KB products by reducing the content of non-nutritional factors to a critical value that exerts a beneficial effect while ensuring that other nutrients are not destroyed is an urgent problem for the future. Based on this, the focus on addressing non-nutritional factors in KBs over the next few years will be in the following key areas: (i) Comprehensive Characterization Studies: Future research goals include fully characterizing the underutilized non-nutritional components in KBs, studying their structure–activity relationships and investigating the correlation between their intake levels and their effects. This related research will lay the foundation for the development of high-value foods, medications and cosmetics driven by the potential of these non-nutritional components as clinical treatments and healthcare applications. (ii) Composite Passivation Method: Many of the existing methods for mitigating non-nutritional factors in KBs rely on a single physical, chemical or biological approach. The judicious selection and combination of multiple methods can significantly enhance the passivation effect, leading to KB products with reduced non-nutritional factors to meet specific dietary needs. Notably, appropriate pretreatment methods can considerably shorten cooking times and enhance KB digestibility. (iii) Molecular Conformational Analysis: Future research will actively explore the impact of the structure of non-nutritional factors on their mechanisms of action and functional effects through spatial conformation analysis and antigenic epitope analysis. This will enable the precise control of quality changes in KBs and their products during KB cultivation and processing. (iv) Genetic Engineering Applications: By gaining an in-depth understanding of the regulatory pathways and genetic manipulation of key non-nutritional factors in plants, researchers will employ metabolomics breeding, infiltration, RNA technology and gene editing to achieve the specific expression of non-nutritional components in seeds while simultaneously reducing non-nutritional factors at their source.

## Figures and Tables

**Figure 1 foods-12-03697-f001:**
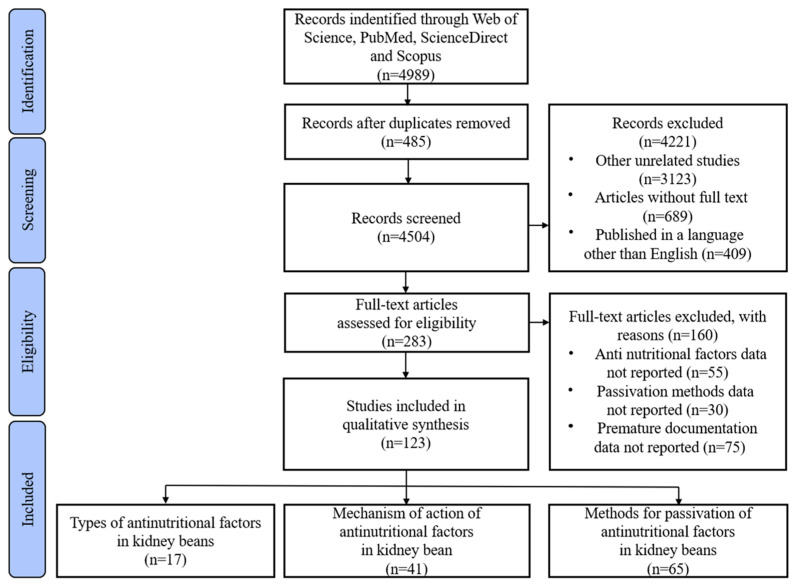
Systematic review flow diagram based on PRISMA guidelines.

**Figure 2 foods-12-03697-f002:**
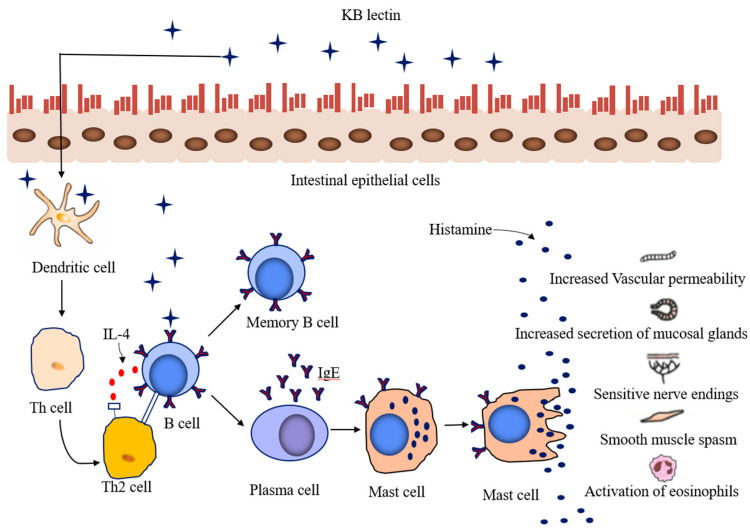
The sensitization mechanism of KB lectin.

**Figure 3 foods-12-03697-f003:**
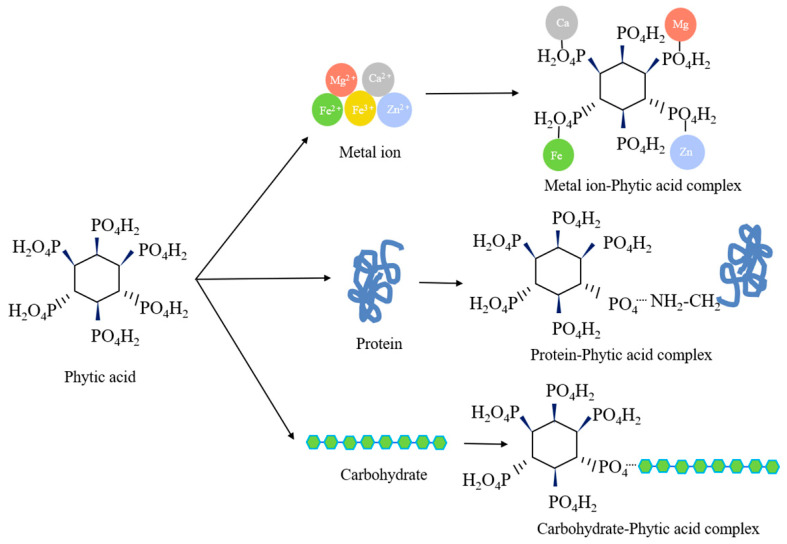
The structure and action mechanism of phytic acid in KB.

**Figure 4 foods-12-03697-f004:**
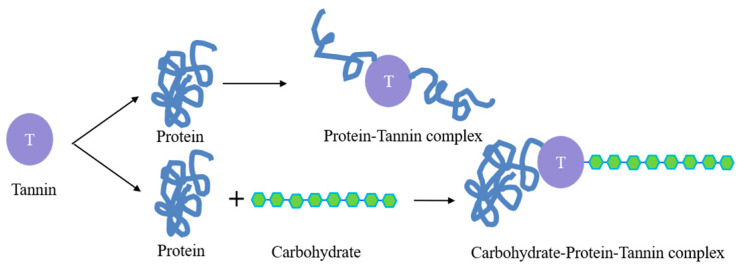
The specific action mechanism of tannins in KBs.

**Figure 5 foods-12-03697-f005:**
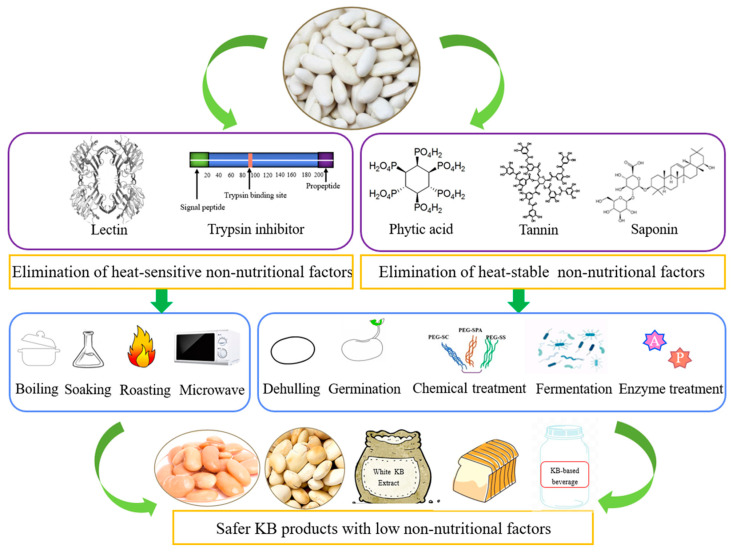
Process for passivating non-nutritional factors in KBs and the development of KB products with low non-nutritional factors.

**Figure 6 foods-12-03697-f006:**
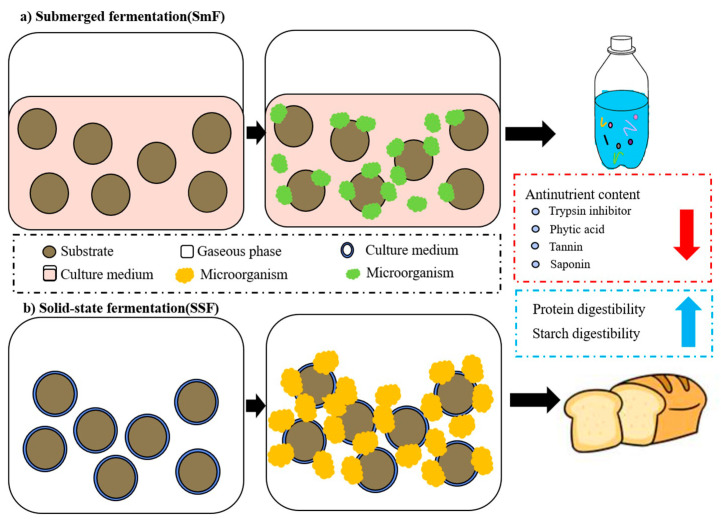
The specific forms of action in submerged and solid-state fermentation processes.

**Table 1 foods-12-03697-t001:** The existing forms and properties of non-nutritional factors in kidney beans (KBs).

Non-Nutritional Factors	Existing Form	Stability Type	Physiological Effects
Lectin	Protein	Heat-sensitive	Protein digestion and absorption [24]
Trypsin inhibitor	Protein	Heat-sensitive	Protein digestion and absorption [25]
Phytic acid	Phytate	Thermal stability	Digestion and utilization of minerals [26]
Tannin	Polyphenol	Thermal stability	Digestion and utilization of carbohydrates [27]
Saponin	Glycoside triterpenoids	Thermal stability	Stimulate the immune system [28]

## Data Availability

Not applicable.

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
