# Peer review of "The Non-Nutritional Factor Types, Mechanisms of Action and Passivation Methods in Food Processing of Kidney Bean (Phaseolus vulgaris L.): A Systematic Review"

_foods, 2023, doi:10.3390/foods12193697_

Round 1

Reviewer 1 Report

* The main issue raised in the work is to demonstrate the presence of various types of antinutritional factors in kidney beans and the mechanisms of their action as well as the ways of their deactivation.

*From the point of view of a scientist and a person potentially interested in the latest trends in food science, this is a very interesting topic. One should remember about the intensively developing market of vegan products. These types of articles are and will be desired by producers, R&D departments and consumers as well as scientists themselves.

* The article introduces the reader in an accessible way to issues related to antinutritional factors, the mechanism of their action and counteracting them.

*The publication is well written. All assumptions contained in the title are addressed in the text. The work is review, so it has a different character from research. Nevertheless, it is a very interesting scientific position.

Dear Authors,

interesting and useful work for other scientists and people interested in the subject. I'd suggest a correction on line 229, it's ".63", it should be "[63]".

Kind regards,

Author Response

Dear Reviewer,

Thank you very much for your correction. We have made changes to line 229 and checked for other similar issues throughout this paper.

Reviewer 2 Report

Dear authors,

While I recognize the value of your work, particularly the way it is written and the image quality, I have some doubts about how it was developed since there is no methodology behind it (you could have used the PRISMA statement or at least identified how the references were found and selected).

I found the manuscript very well structured and comprehensible; however, without a structured methodology, I think it is directed to the general public, furthermore considering the lack of novelty for the scientific community.

Author Response

Dear Reviewer,

Thank you very much for your valuable comments. We have revised the article using PRISMA guidelines to ensure a structured methodology. In addition, we have added some references, added Figure 5, and revised the conclusion and prospective section. We have made extensive modifications and additions to the manuscript to make it more suitable for publication in professional journals.

Reviewer 3 Report

The manuscript's subject “Antinutritional factors types, mechanism of action and passivation methods in food processing”, has so far been frequently and extensively discussed by many authors. Therefore, in my opinion, the scientific level of the presented material is unsatisfactory for a review manuscript. The work touches on a great many aspects and themes, but they are discussed very generally and, in my opinion, too briefly. The various subsections actually point out information without any deeper explanation. In my opinion, referring to a single source of literature when discussing such a popular topic and in a review article is absolutely insufficient. Reading the manuscript most of the time I felt like reading an academic textbook.

In my opinion, the article needs to be rewritten and adequately developed and supported by numerous literature sources of the presented content.

Author Response

Dear Reviewer,

Thank you very much for your valuable comments. We have revised the article using PRISMA guidelines to ensure a structured methodology. In response to the lack of references, we have added some references in the manuscript and added more in-depth discussions. In addition, we have added Figure 5 to provide a more intuitive display of the digestion method of non-nutrient components in kidney beans. We have significantly revised the conclusion and prospective section, hoping to provide more useful and in-depth information to researchers in related professions. We have made extensive modifications and additions to the manuscript to make it more suitable for publication in professional journals.

Round 2

Reviewer 2 Report

Dear authors,

Your paper improved a lot with the introduction of the methods section. In my opinion, it should be accepted after some modifications that I identified as major just to ensure I can reread the paper before approval.

I noticed you changed the therm antinutritional to non-nutritional. I understand why, as the "antinutritional" factors can have positive effects when used in other applications. However, I think you should maintain the term antinutritional, explaining in which sense you are using it in the introduction. In my opinion non-nutritional stands for something that is not nutritional at all, however, when you supplement a patient with one of these non-nutritional they become a functional nutrient. I know the use of the term is controversial, and therefore this discussion needs to be present in your paper since it is the main subject. Maybe there is any reference you can add to explain this.

I also noticed there is no reference to the "kidney bean" species. In the introduction, it should be clear what are we talking about, in this case, kidney beans. What are kidney beans specifically? Do you refer to any specific sun species or it is just Phaseolus vulgaris?

Additionally, I think Kidney beans can be referred to as KB in the text since it is repeated throughout all the manuscript (I look at the paper and see a LOT of Kidney beans, Kidney beans, Kidney beans, Kidney beans!)

You included the methods section, however, it is not explicit in the abstract. In my opinion, you should add a line explaining you used the PRISMA and the period of research reviewed.

L 30 Those nutrients explain the popularity of beans, however at first glance it seems you are talking about the main components of beans and omitting starch. Please rephrase for clarity.

L 32 missing reference

L 51 and 56 I would not use "etc" but "among others"

Fig 1 If you did not use other sources you can delete the box. The records excluded in the screening phase should be explained as "(factor x,y,z...)", you don't need to specify the numbers for each factor, but the number of exclusions is huge, therefore in my opinion it should be clarified in the scheme. You can also align the figure blocks.

L 140-141 repeated sentence

Table 1 what do you mean by utilisation? utilization by the body? maybe consider other words

L 251 Please remove "the mineral"

In the results section, in the antinutritional factors part, I suggest you make a table with the content of each antinutritional factor: description of the substance, mechanism of action, problem caused in the body, other uses for the substance, and then reformulate each section according to this structure. I notice in some cases the potential benefits appear at the end of the section and in other cases at the beginning.

L 428 Please do not use the word milk, instead use "beverage" or "kidney bean-based beverage"... milk is produced by mammals!

514 PHA? If you want to use abbreviations please make sure they are explained the first time they appear in the text

L 517 GI is not needed unless it would be used later (and I think the full expression was used before without mentioning the abbreviation)

L 530  Please use  “(or submerged fermentation)” instead of "and liquid-state fermentation is also called submerged fermentation"

L 575 missing reference

L 594 how do you know what will be the new key areas?

I think this is an editorial question, but the DOI is missing in the references. Was that intentional?

Good work!

Author Response

Dear Reviewer,

We appreciate your valuable comments very much, which are helpful to improve the quality of our works. Thank you very much for your patience. We carefully modified this article according to the comments given by the reviewers.

According to the comments, we have revised our paper as follows:

Comment 1 from Reviewer: I noticed you changed the therm antinutritional to non-nutritional. I understand why, as the "antinutritional" factors can have positive effects when used in other applications. However, I think you should maintain the term antinutritional, explaining in which sense you are using it in the introduction. In my opinion non-nutritional stands for something that is not nutritional at all, however, when you supplement a patient with one of these non-nutritional they become a functional nutrient. I know the use of the term is controversial, and therefore this discussion needs to be present in your paper since it is the main subject. Maybe there is any reference you can add to explain this.

Response: In fact, this modification was made based on the editor’s comments. Based on your suggestion, we have expanded upon the multi-faceted effects of non-nutritional factors on human health in the second paragraph of the Introduction section and included some relevant references.

Comment 2 from Reviewer: I also noticed there is no reference to the "kidney bean" species. In the introduction, it should be clear what are we talking about, in this case, kidney beans. What are kidney beans specifically? Do you refer to any specific sun species or it is just Phaseolus vulgaris?

Response: Thank you for your valuable comments. The Kidney Bean mentioned in this article refers specifically to Phaseolus vulgaris L., and does not include any other beans. We have made this clear in the first paragraph of the Introduction section.

Comment 3 from Reviewer: Additionally, I think Kidney beans can be referred to as KB in the text since it is repeated throughout all the manuscript (I look at the paper and see a LOT of Kidney beans, Kidney beans, Kidney beans, Kidney beans!)

Response: Thank you for your valuable comments, and the modifications have been made according to your suggestions.

Comment 4 from Reviewer: You included the methods section, however, it is not explicit in the abstract. In my opinion, you should add a line explaining you used the PRISMA and the period of research reviewed.

Response: Thank you for your valuable comments, and the modifications have been made according to your comments.

Comment 5 from Reviewer: L 30 Those nutrients explain the popularity of beans, however at first glance it seems you are talking about the main components of beans and omitting starch. Please rephrase for clarity.

Response: Thank you for your valuable comments, and the modifications have been made according to your comments.

Comment 6 from Reviewer: L 32 missing reference

Response: Thank you for your valuable comments, and the modifications have been made according to your comments.

Comment 7 from Reviewer: L 51 and 56 I would not use "etc" but "among others"

Response: Thank you for your valuable comments, and the modifications have been made according to your comments.

Comment 8 from Reviewer: Fig 1 If you did not use other sources you can delete the box. The records excluded in the screening phase should be explained as "(factor x,y,z...)", you don't need to specify the numbers for each factor, but the number of exclusions is huge, therefore in my opinion it should be clarified in the scheme. You can also align the figure blocks.

Response: We have made the necessary modifications to the expressions in Figure 1 and Section 3.1, ensuring clarity and accuracy. In addition, we have also modified the format of Figure 1.

Comment 8 from Reviewer: L 140-141 repeated sentence

Response: Thank you for your valuable comments, and the modifications have been made according to your comments.

Comment 9 from Reviewer: Table 1 what do you mean by utilisation? utilization by the body? maybe consider other words

Response: Thank you for your valuable comments, and we have replaced 'utilisation' with 'absorption'.

Comment 10 from Reviewer: L 251 Please remove "the mineral"

Response: Thank you for your valuable comments, and the modifications have been made according to your comments.

Comment 11 from Reviewer: In the results section, in the antinutritional factors part, I suggest you make a table with the content of each antinutritional factor: description of the substance, mechanism of action, problem caused in the body, other uses for the substance, and then reformulate each section according to this structure. I notice in some cases the potential benefits appear at the end of the section and in other cases at the beginning.

Response: Regarding Table 1, while it serves this intended purpose, it lacks certain content you mentioned, such as the content on non-nutritional factors. Due to significant variations in purification methods across different references, comparing such content may not hold scientific significance. The benefits of various non-nutritional factors are not the central focus of this article, so we have included them within the corresponding sections and have also added some references for them in this revised version. The benefits of various non-nutritional factors are placed at the end of the corresponding sections as you suggested.

Comment 12 from Reviewer: L 428 Please do not use the word milk, instead use "beverage" or "kidney bean-based beverage"... milk is produced by mammals!

Response: Thank you for your valuable comments, and the modifications have been made according to your comments.

Comment 13 from Reviewer: L 514 PHA? If you want to use abbreviations please make sure they are explained the first time they appear in the text

Response: Thank you for your valuable comments, and the modifications have been made according to your comments.

Comment 14 from Reviewer: L 517 GI is not needed unless it would be used later (and I think the full expression was used before without mentioning the abbreviation)

Response: Thank you for your valuable comments, and the modifications have been made according to your comments.

Comment 15 from Reviewer: L 530  Please use  “(or submerged fermentation)” instead of "and liquid-state fermentation is also called submerged fermentation"

Response: Thank you for your valuable comments, and the modifications have been made according to your comments.

Comment 16 from Reviewer: L 575 missing reference

Response: Since generally speaking there is no need to add references to the Conclusion section, no modifications were made here.

Comment 17 from Reviewer: L 594 how do you know what will be the new key areas?

Response: In the Conclusion section, we have made modifications and added some explanations. It's worth noting that many review articles provide predictions about key areas at Conclusion section. These predictions, while somewhat subjective, are based on judgments formed through an extensive review of the cited references.

Comment 18 from Reviewer: I think this is an editorial question, but the DOI is missing in the references. Was that intentional?

Response: Thank you for your valuable comments, and the modifications have been made according to your comments.

All the changes in the article have been marked with red font colors. Thank you very much for your valuable time. If you have any question, please contact us without any hesitate.

Best Regards,

Sincerely yours,

Tiezheng MA

Reviewer 3 Report

The authors have made a number of significant changes. 

Author Response

Dear Reviewer,

We appreciate your valuable comments very much, which are helpful to improve the quality of our works. Thank you very much for your patience. We carefully modified this article according to the comments given by the reviewers and editors.

We have substantially increased the number of cited literature sources to reinforce the arguments and viewpoints presented in the article. This expanded range of literature citations, sourced from multiple perspectives and authoritative references, enhances the authority and credibility of our review.

To align with the standards of a systematic review, we have incorporated PRISMA Flow Diagrams and restructured the framework of the article. These changes enhance the logical organization of the manuscript.

In response to the editor's comments, we have replaced "anti-nutritional factors" with "non-nutritional factors" throughout the manuscript. Additionally, we have provided extensive explanations in the second paragraph of the Introduction section to elucidate the multifaceted effects of non-nutritional factors on human health, accompanied by relevant supporting literature.

Our manuscript could fill a unique gap in the field, since there have been no review articles with the similar title in the past several years. We believe that this article will provide researchers and practitioners in related fields with valuable, in-depth information, contributing to the advancement of this specific area of research.

We have enriched the Conclusion section by integrating insights from the extensive references cited in our review. This allows us to make comprehensive predictions regarding the future development trends in the key areas discussed in our manuscript.

Throughout various sections of the article, we have engaged in more comprehensive discussions and provided detailed explanations, particularly concerning the types, mechanisms of action, and methods of digestion of non-nutritional factors. We have augmented these discussions with additional new reference citations to assist researchers and practitioners in gaining a deeper understanding of these concepts.

To enhance the clarity and comprehensibility of the article, we have added new figure that visually illustrate and support our key points. These visual aids are intended to facilitate a better understanding of complex concepts and engage the interest of researchers and practitioners in related fields.

We appreciate your thorough review of our manuscript, and we believe that these revisions significantly strengthen the quality and impact of our work. All the changes in the article have been marked with red font colors. Thank you very much for your valuable time. If you have any question, please contact us without any hesitate.

Best Regards,

Sincerely yours,

Tiezheng MA